# Model of Genetic Code Structure Evolution under Various Types of Codon Reading

**DOI:** 10.3390/ijms23031690

**Published:** 2022-02-01

**Authors:** Paweł Błażej, Konrad Pawlak, Dorota Mackiewicz, Paweł Mackiewicz

**Affiliations:** Department of Bioinformatics and Genomics, Faculty of Biotechnology, University of Wrocław, ul. Joliot-Curie 14a, 50-383 Wrocław, Poland; konrad.pawlak@uwr.edu.pl (K.P.); dorota.mackiewicz@uwr.edu.pl (D.M.); pamac@smorfland.uni.wroc.pl (P.M.)

**Keywords:** amino acid, codon, evolution, genetic code

## Abstract

The standard genetic code (SGC) is a set of rules according to which 64 codons are assigned to 20 canonical amino acids and stop coding signal. As a consequence, the SGC is redundant because there is a greater number of codons than the number of encoded labels. This redundancy implies the existence of codons that encode the same genetic information. The size and organization of such synonymous codon blocks are important characteristics of the SGC structure whose evolution is still unclear. Therefore, we studied possible evolutionary mechanisms of the codon block structure. We conducted computer simulations assuming that coding systems at early stages of the SGC evolution were sets of ambiguous codon assignments with high entropy. We included three types of reading systems characterized by different inaccuracy and pattern of codon recognition. In contrast to the previous study, we allowed for evolution of the reading systems and their competition. The simulations performed under minimization of translational errors and reduction of coding ambiguity produced the coding system resistant to these errors. The reading system similar to that present in the SGC dominated the others very quickly. The survived system was also characterized by low entropy and possessed properties similar to that in the SGC. Our simulation show that the unambiguous SGC could emerged from a code with a lower level of ambiguity and the number of tRNAs increased during the evolution.

## 1. Introduction

The structure and properties of the standard genetic (SGC) code have intrigued scientists since the first ascriptions of codons to amino acids were discovered in the sixties of the last century [1,2]. From that time, many hypotheses concerning the origin and evolution of SGC have been proposed (see for review: [3,4,5,6,7,8]). However, it is still unclear which factors played a decisive role in the process of the genetic code emergence and evolution.

These investigations been started with two of Crick’s seminal papers, i.e., [9,10]. In the first work, the author discussed the general codon block structure of the SGC. Briefly, the codons in the SGC are arranged in groups, which is an immediate consequence of the fact that the number of possible codons, i.e., 64, is greater than the number of encoded items, i.e., 20 amino acids and a stop translation signal. The codon groups encoding the same item are called synonymous and are composed of two, three, four, or six codons. Codons in a given group differ mostly in the third codon position. In order to explain this fact, Crick proposed the wobble rule, which refers to interactions between the first base in a tRNA anticodon and the third base of translated codon in a transcript (mRNA). This rule assumes that the base pairing between these two RNAs does not have to follow Watson–Crick base pair rules, i.e., cytosine–guanine and adenine–uracil, but that other interactions are also possible, i.e., guanine–uracil, hypoxanthine–uracil, hypoxanthine–adenine and hypoxanthine–cytosine. Additional experiments revealed that other modified bases can also pair with the typical ones [11].

This phenomenon has fundamental consequences. It reduces the number of necessary tRNA molecules for protein synthesis. Moreover, it determines the structure of codon blocks and introduces specific robustness of the SGC against single nucleotide substitutions occurred in the third position in codons belonging to a group encoding the same genetic information. Therefore, such mutations do not change coded amino acids.

In the second paper [10], Crick considered the evolution of genetic code on a general level. He put forward “the frozen accident” hypothesis. According to this scenario, it is not inconceivable that, in the past, there were various genetic codes that were used by different organisms. They coexisted and evolved simultaneously. Finally, the present structure of the SGC won this competition by accident and stayed universal among all domains of life, because any change in the codon assignment would be highly deleterious at later stages of SGC evolution.

The existence of many different coding systems at the early evolution of genetic code seems very probable. This assumption appears to be a good starting point for further investigations of the SGC evolution. It was proposed that these codes evolved together from a set of ambiguous codon assignments towards a coding system with a low level of ambiguity to reduce an initial high translational noise [5,12,13,14]. During the genetic code evolution, amino acids were gradually added to the code, which was profitable because it increased the diversity of synthesized proteins [6,15,16,17].

These assumptions were included in a simulation model to show the origin of coding system structure [13,14]. This model assumed that every code was a set of rules, described by selected random variables, according to which 64 codons were assigned to specific genetic information, i.e., labels. What is more, this model did not take into account any properties of amino acids in order to avoid additional assumptions, which made the model general. The model considered three types of reading mechanisms, called $M_{1}$, $M_{2}$, and $M_{3}$, which induced different codon groups assigned to respective encoded labels. In $M_{1}$, a given amino acid was coded by codons that had two fixed identical positions and differed in one position from the reference codon. $M_{2}$ assumed that a given amino acid was coded by codons with one fixed identical position and differed in exactly one of the other two codon positions from the reference codon. Codons of $M_{3}$ for a given amino acid differed in exactly one of any codon positions from the reference codon. Thereby, the codon groups defined by the $M_{1}$ rule had a structure similar to that in the SGC, whereas $M_{2}$ and $M_{3}$ were potential generalizations of $M_{1}$. The codes were evolved to a state characterized by a low level of uncertainty.

The results based on these simulations showed that the structure of the SGC would emerge from highly ambiguous coding systems under relatively simple criteria, i.e., the reduction of translational noise and a stepwise addition of amino acids into the code [13,14]. The codon block structure observed in the best surviving code was characterized by low translational noise as well as a high robustness against point mutations. However, the authors assumed that the type of reading, i.e., $M_{1}$, $M_{2}$, and $M_{3}$, stayed fixed and constant during simulations. Therefore, to make the simulation model more realistic, we assumed here that these three mechanisms of reading genetic information could coexisted at the same time and evolved simultaneously according to Crick’s scenario. The computer simulations performed including this possibility showed that the initial coding systems evolved very quickly to one characterized by unambiguous reading of genetic information.

## 2. Results

### 2.1. Changes of Genetic Codes during Simulations

We performed computer simulations to study the process of coding system emergence. Each simulation was run up to 100,000 steps. All simulations started with a population of 1000 genetic codes. They were described by initially randomly generated probability distribution functions for codon assignments and reading system types. During the simulations, we collected several descriptive characteristics for a given coding system, i.e., values of the fitness function, the expected number of genetic labels encoded by a given reading mechanism, and the structure of codon blocks. They together allowed us to describe tendencies in the evolution of coding systems. All simulations were run under different seeds because we wanted to find out if the observed tendencies in the genetic code structure have a general character, i.e., is they are independent of starting constraints.

The simulations started from non-optimized genetic systems. However, the fitness values showed a tendency to increase substantially. During the code evolution, the fitness function increased from a low value to higher one after 15,000 steps. After some fluctuations it stabilized after 50,000 steps above the value of −20 in the logarithmic scale (Figure 1).

The increase in the fitness function corresponds to the decrease in entropy values calculated for probability distributions of codon assignments and reading types. The average genetic code entropy decreased with the simulation time and remained rather constant at a low level after 30,000 steps (Figure 2A). However, the drop and stabilization of the average entropy for reading systems occurred much earlier, i.e., after 5000 steps (Figure 2B). The results show that the studied coding systems changed their properties during simulation runs. However, the reading system was selected by the genetic codes much faster than the translational noise was reduced.

### 2.2. Structure of Genetic Codes

Interestingly, we observed a relatively fast emergence of nearly homogeneous reading systems, i.e., one type out of $M_{1}$, $M_{2}$, and $M_{3}$ was preferred for all possible labels l=1,2,3,…21. The model $M_{1}$ started to dominate very quickly over the two others (Figure 3). 

This tendency was detected from the beginning of simulations. The $M_{3}$ system was the first one eliminated after about 2500 steps and $M_{2}$ coexisted with $M_{1}$ to about 20,000 steps. After that, only $M_{1}$ was used by the genetic codes to read encoded information, whereas the other systems, i.e., $M_{2}$ and $M_{3}$, disappeared and the coding system stayed homogeneous. The homogeneity is visualized by the heatmap, which shows that $M_{1}$ was selected as the only reading type receiving the probability one for all encoded labels (Figure 4). 

In contrast to that, no reading system was preferred at the beginning of simulations (Figure 5).

It is also interesting to investigate the codon block structure of the computed coding system at the end of simulations. The results presented in in the previous paragraph indicate that the $M_{1}$ reading system was the most preferred, which suggests that codon blocks induced by this rule should be present in the structure of the genetic codes selected after simulations. In Figure 6, we present the coding system that was characterized by the highest fitness value among all simulation runs. In contrast to the beginning of simulations (Figure 7), we can distinguish for each label a group of codons with a high coding probability, i.e., over 0.8.

Many of these groups were composed of codons that differed in one codon position as assumed by the $M_{1}$ rule and resemble the structure of the standard genetic code.

The distribution of size of codon groups encoding a given label is also very interesting (Figure 8). Eleven labels were encoded by blocks consisting of four codons, eight labels by two-codon blocks, one label by three-codon blocks, and one label by one codon. This is very similar to the number of codon blocks in the SGC, in which five amino acids are encoded by four-codon blocks, nine amino acids by two-codon blocks, one amino acid and the stop signal by three-codon blocks, and two amino acids by one codon each. Moreover, there are three amino acids, each of which is ascribed to six codons. These codons are, however, organized into the subgroup of four codons and the subgroup of two codons. These tendencies were observed in all genetic codes evolved at the end of all simulation runs.

## 3. Discussion

In this paper, we investigated a potential emergence of genetic codes from a set of highly ambiguous codon assignments to the codes characterized by an unequivocal assignment of codons to encoded labels. A similar problem was discussed in our previous papers [13,14], where we showed that the structure of genetic codes similar to that in the SGC could evolve under error minimization restrictions and the stepwise addition of amino acids to the genetic code. However, in the previous studies, the method of reading genetic information and its degeneracy, or more precisely, the way of recognizing not only a codon but also its codon neighborhood encoding the same label, was fixed and unchanged from the start of genetic code evolution. Therefore, in this approach, we not only included the evolution of codon assignments, but also took into account the evolution of the reading systems, which are responsible for codon block structures in the genetic code.

This assumption seems reasonable because the genetic code at very early stages of evolution was likely a mixture of different reading systems for the set of encoded items and characterized by ambiguity in the coding of amino acids until the whole system became more precise [5,12]. The genetic code could have evolved from a highly ambiguous and non-homogeneous system to an unambiguous system with nearly identical reading mechanisms for each encoded label. The results of our simulations are in agreement with the 2-1-3 model [18,19] and the four-column theory [15], which also assume that the degeneracy of the genetic code was subsequently reduced with time. Initially, the second codon position was used to determine encoded amino acids, whereas other codon positions were meaningless. Only later the additional codon position positions were used for distinguishing of coded amino acids.

In our simulations, we tested three different models of reading systems, namely, $M_{1}$, $M_{2}$, and $M_{3}$, which from the beginning of simulations could be used to read genetic information simultaneously with different probabilities. The model $M_{3}$ was the most tolerant in codon reading because it assumed that nine other codons can code for the same label as the reference codon. In $M_{2}$, six codons could have the same meaning, whereas in $M_{1}$, three codons besides the reference codon could code for the same label.

Although the simulations produced coding systems with high unambiguity in reading genetic information, there are some codons that can still code more than one label, one with a very high and the other with a very lower probability (Figure 6). This resembles the situation in natural biological systems because the present translational machinery also shows errors with rates of $10^{-3}$ to $10^{-6}$ per codon [20] or $10^{-3}$ to $10^{-5}$ per incorporated amino acid [21,22,23,24]. Mistranslation errors can be beneficial for parasites, for example, in adaptation to oxidative and environmental stresses as well as in host invasion and evasion of host immunity [25,26,27,28,29].

The evolution of ambiguous codes to those showing unequivocal assignment of codons implies that the number of tRNAs matching appropriate codons in protein synthesis increased during evolution. In the initial state of high ambiguity, a small fraction of tRNA could be enough to recognize codons due to the tolerant codon reading. When the codes became more unequivocal, a larger number of tRNAs were required to distinguish the codons and assign an appropriate amino acid. This scenario corresponds well to the view that the number of genes coding for tRNAs and aminoacyl-tRNA synthetases charging amino acids to the tRNA molecules increased via duplication during the genetic code evolution [7,16,30,31,32,33]. The increase in the number of tRNAs is associated with the more precise reading of genetic information.

The population of genetic codes evolved to reduce coding errors, which was manifested by a higher probability that a given codon encodes a fixed label. What is more, we also observed that the model $M_{1}$ started to dominate in coding systems among all possible types of reading very quickly. Therefore, at the end of simulations. we obtained genetic codes with low entropy values and a nearly homogeneous reading system. It is worth mentioning that the $M_{1}$ rule generates codon blocks with similar properties to those observed in the standard genetic code. Moreover, the codes at the end of the simulations were similar to the SGC in the number and structure of codon blocks encoding the same label. Such organization means that the codes are robust to point mutations that could change the encoded information, because changing a nucleotide in one codon position does not result in changing the encoded label.

This tendency to minimize errors in the coded amino acid replacements is also present in the SGC and has been reported in many analytical and statistical studies [34,35,36,37,38,39,40,41,42,43]. However, the minimization of the errors turned out to be imperfect when genetic algorithms were applied and the SGC was compared with optimized codes [13,19,44,45,46,47,48,49,50]. Interestingly, alternative codes appeared to better minimize the harmful effects of mutations than the SGC [51,52]. This means that there are possibilities for improving the SGC. However, substantial changes in the code would be deleterious due to the universality of this code associated with beneficial exchange of genetic information between organisms, e.g., via horizontal gene transfer [16]. Therefore, other mechanisms, e.g., optimization of mutational pressure, evolved to mitigate the mutation errors associated with replication [53,54,55,56].

## 4. Materials and Methods

### 4.1. Representation of Genetic Codes

In order to investigate the problem of genetic structure evolution, we consider a population of 1000 theoretical genetic codes, i.e., candidate solutions, where each coding system is represented as a matrix $P=(p_{cl}),\;1\leq c\leq 64$ and 1≤l≤21 with 64 rows, i.e., codons *c*, and 21 columns, i.e., amino acids and the stop translation signal, or more generally, encoded labels or items *l*. Each row in the matrix P denotes a probability distribution function of codon assignments to one out of 21 possible labels. Therefore, $p_{cl}$ is the probability that a codon *c* encodes a label *l*. As a result, P describes a potential ambiguity in codon ascription. Figure 6 is a graphical representation of the coding system at the beginning of the simulations. The color gradient represents the probability of coding a given label in a column by a given codon in a row.

This approach has been used in our previous works [13,14], where we studied the process of genetic code evolution from a set of ambiguous codon assignments characterized by a large value of entropy to nearly unambiguous coding systems. However, it was assumed that the type of reading system was fixed during the whole simulation run. By a reading system, we mean the type of inaccuracy in codon recognition. In order to explain this important feature, let us consider that a codon *c* has the highest probability to encode a given label *l*. We also assume that the reading system allows that several codons belonging to the neighborhood of the reference codon *c* can encode the same label *l*. We considered in simulations the following different ways of reading:*M*_1_ all codons belonging to a given group encoding a fixed label have two fixed codon positions identical with codon *c* and differ in exactly one nucleotide at other codon positions;*M*_2_ all codons belonging to a given group encoding a fixed label have one fixed codon position with codon *c* and differ in exactly one nucleotide in one of the other two codon positions;*M*_3_ all codons belonging to a given group encoding a fixed label differ in exactly one nucleotide with codon *c* in any codon position.

For example, let us assume that codon GGG encodes a fixed label *l*. Then the neighborhood of GGG is:GGG,GGA,GGC,GGT for the model $M_{1}$;GGG,AGG,CGG,TGG,GAG,GCG,GTG for the model $M_{2}$;GGG,AGG,CGG,TGG,GAG,GCG,GTG,GGA,GGC,GGT for the model $M_{3}$.

### 4.2. Simulation Procedure

In contrast to [13,14], we took into account the fact that the reading systems can evolve at the same time and compete with each other. In other words, the reading mechanisms $M_{1},M_{2}$, and $M_{3}$ can coexist in one coding system to encode the same labels with different probabilities. Thus, we considered that the respective reading mechanism for a given encoded label is a random variable. This variable was generated at the beginning of simulated evolution. Figure 5 shows probability distribution functions of these three reading systems (y-axis) for each encoded label (x-axis). The probability values are depicted by respective color gradients. As we can see, the reading system is highly ambiguous, because a given label can be read by different mechanisms with various probabilities.

To sum up, we had two parameters responsible for shaping the structure of genetic code in our simulations. The first one is described by the probability that a given codon encodes a selected label. The second parameter is described by the probability that a given genetic label is read by a mechanism $M_{1},M_{2}$, and $M_{3}$.

We ran simulations using the methodology of evolutionary algorithms. This was applied to find solutions in optimization problems where it is not possible to use classical analytical methods because their respective assumptions do not hold. According to this approach, the simulation process is divided into steps called generations. During each step, a population of individuals (solutions) is subjected to two important operators, namely, mutation and selection. They both act on the potential solutions, but in different way. The mutation is responsible for maintaining diversity of population, so we can test many different solutions during one generation. Thanks to the selection, it is possible to choose better solutions in terms of fitness. Such individuals have a greater probability to reproduce.

In the case considered here, the mutation operator randomly modifies the probability distribution functions for the assignment of codons and the type of reading system. The selection operator chooses better solutions with a greater probability for the next generation. Therefore, the central role in the simulations is the fitness function *F*, which allowed us to measure the quality of evolving genetic codes.

### 4.3. The Fitness Function F

Simply speaking, *F* is a modified version of the fitness function used in [13,14]. Similarly to [13], for each label l=1,2,…,21, we choose a respective codon $c_{i}$ according to the Bayes rule. Therefore, we obtain the sequence of codons $C=c_{1},c_{2},\ldots c_{21}$, which is the most probable path to encode all 21 labels. What is more, each codon $c_{i}$ belonging to *C* has its own neighborhood $N_{j}\left(c_{i}\right)$, describing a potential *j* way of reading. We considered j=1,2,3 three types of neighborhoods defined according to the $M_{1},M_{2}$, and $M_{3}$ reading models. In contrast to the previous works, the type of codon neighborhood was assigned at random to each codon $c_{i}\in C$. This fact was manifested in the form of the fitness function *F*, which is defined in the following way:$$F=\sum_{c_{1}^{\prime }\in N_{j_{1}}\left(c_{1}\right),\ldots ,c_{21}^{\prime }\in N_{j_{21}}\left(c_{21}\right)}\frac{P\left(l=1|c_{1}^{\prime }\right)P\left(j_{1}\right)}{|N_{j_{1}}\left(c_{1}\right)|}\frac{P\left(l=2|c_{2}^{\prime }\right)P\left(j_{2}\right)}{|N_{j_{2}}\left(c_{2}\right)|}\cdot \ldots \cdot \frac{P\left(l=21|c_{21}^{\prime }\right)P\left(j_{21}\right)}{|N_{j_{21}}\left(c_{21}\right)|}\;,$$
where $P\left(j_{i}\right),\;j_{i}=1,2,3$ is a probability that a label *i* is encoded by a codon group $N_{j_{i}}\left(c_{i}\right)$, which is defined according to reading the $M_{j_{i}}$ constraint.

For a fixed label *l*, the formula
$$\frac{P\left(l|c^{\prime }\right)P\left(j\right)}{|N_{j}\left(c\right)|},\;c^{\prime }\in N_{j}\left(c\right)$$
has a simple interpretation. In the denominator, there is a joint probability that the codon neighborhood $N_{j}\left(c\right)$ of reading type $M_{j}$ is chosen and the codon $c^{\prime }\in N_{j}\left(c\right)$ encodes the label *l*. This probability is normalized by the size of codon block $N_{j}\left(c\right)$. Thus, the fitness function *F* gives information about the coding strength of genetic code. It is calculated over all possible combinations of codons $c_{1}^{\prime },c_{2}^{\prime },\ldots c_{21}^{\prime }$ belonging to randomly selected codon neighborhoods.

In Figure 1, we presented the general tendency of the fitness function observed during simulations. The plot presents 10 simulation runs with different initial seeds. The values of the fitness function are shown in the logarithmic scale. We applied the general additive model (GAM) approximation method to illustrate the changes in this function during the simulations. As we can see, the fitness values increase from very low values to very high ones after 20,000 generations and then stabilize. This indicates that the simulations converged to very similar final results independently of the initial state. Moreover, the course of the curves means that the studied genetic code evolved from that characterized by a high ambiguity to that showing a low ambiguity in the assignment of codons to labels and the selection of reading mechanisms.

### 4.4. Measure of the Quality of Genetic Codes

In order to describe the structural properties of the genetic codes represented as the matrix $P=(p_{cl})$, we used the entropy
$$H_{c}\left(P\right)=-\sum_{c=1}^{64}\sum_{l=1}^{21}p_{cl}log\left(p_{cl}\right),$$
which is the sum of Shannon entropy calculated for each row of the matrix P over the probabilities that a codon *c* encodes a label *l*. Higher values of entropy indicate that a given coding system is composed of ambiguous assignments of codons to labels, whereas lower values mean that the coding is unambiguous.

Similarly, for the entropy of codon to label assignments, we calculated the entropy for the distributions of three reading systems.
$$H_{r}\left(P\right)=-\sum_{l=1}^{21}\sum_{j=1}^{3}p\left(j\right)log\left(p\left(j\right)\right),$$
where *l* is an encoded label, *j* is a type of reading, and p(j) is a probability that *j*-type was chosen for reading a given label.

## Figures and Tables

**Figure 1 ijms-23-01690-f001:**
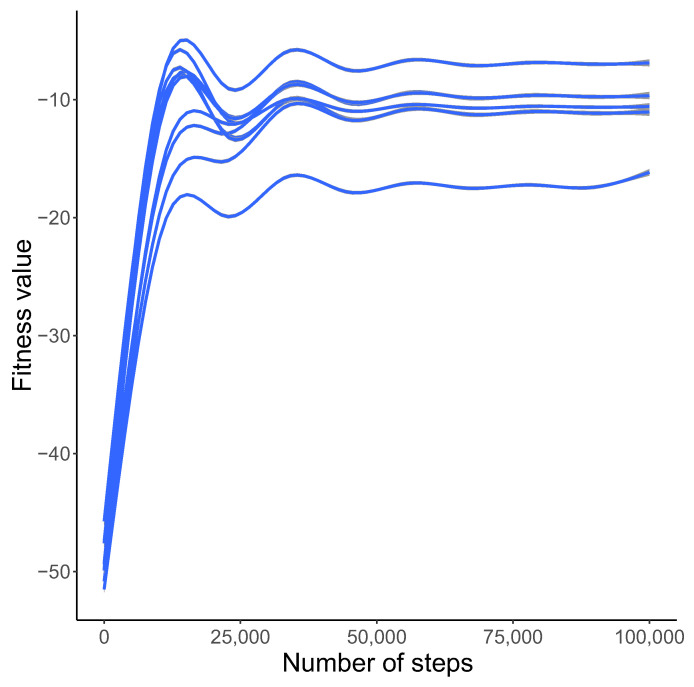
Changes in the best approximation of the fitness function *F* (y-axis) with the number of generations (x-axis) based on the GAM model and 10 simulation runs with different initial seeds. The y-axis is shown in a logarithmic scale.

**Figure 2 ijms-23-01690-f002:**
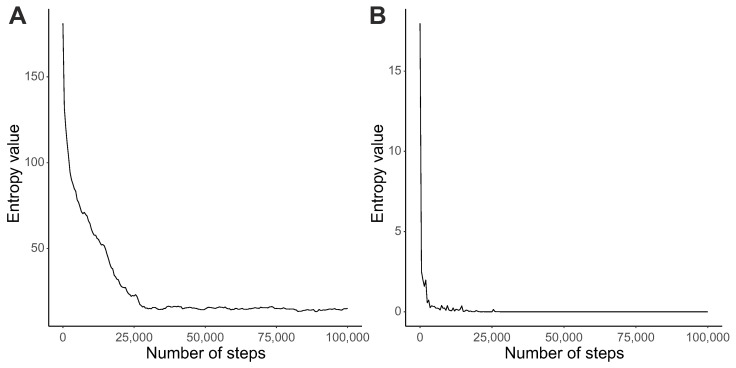
Changes in the average entropy values (y-axis) calculated from distributions of codon assignments $H_{c}\left(P\right)$ (**A**), and reading system types $H_{r}\left(P\right)$ (**B**), with the number of generations (x-axis).

**Figure 3 ijms-23-01690-f003:**
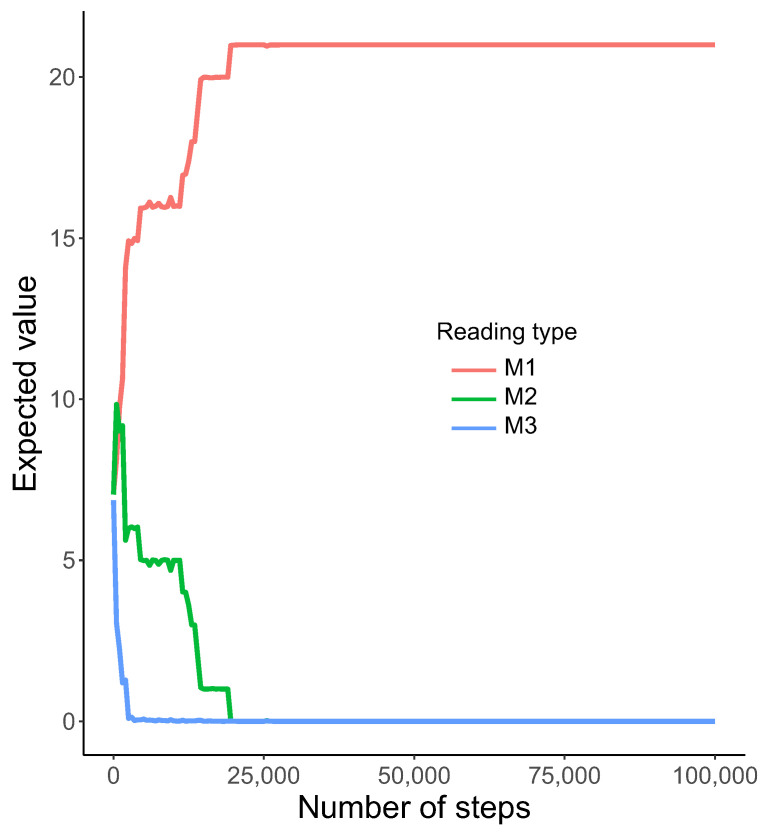
The expected value of the total number of encoded labels using various reading types $M_{1}$, $M_{2}$, and $M_{3}$ (y-axis) with the number of generations (x-axis). Notice that the expected number of labels read by the $M_{1}$ system started dominated among all considered types of reading very quickly.

**Figure 4 ijms-23-01690-f004:**
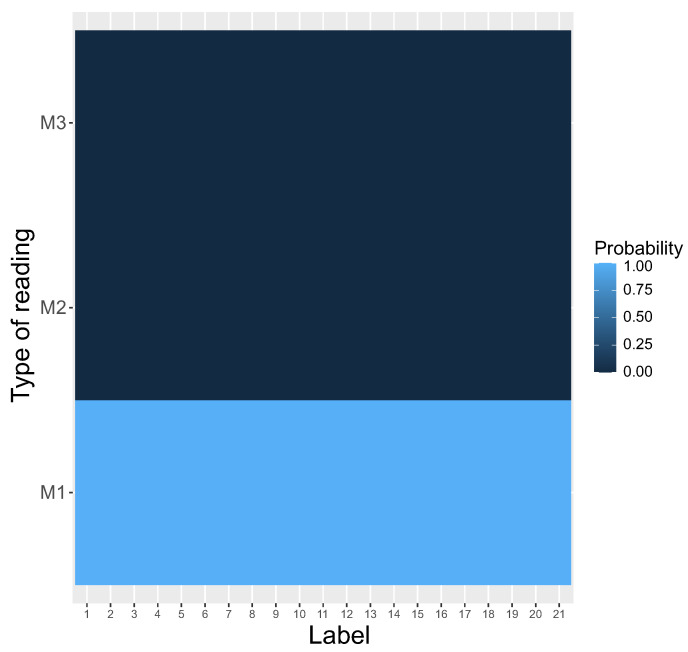
The heatmap of different types of reading systems (y-axis) at the end of the simulations. This is a graphical representation of a matrix in which each genetic label (x-axis) ascribes a probability that is read by a given reading type. Please compare with Figure 5 at the beginning of the simulations.

**Figure 5 ijms-23-01690-f005:**
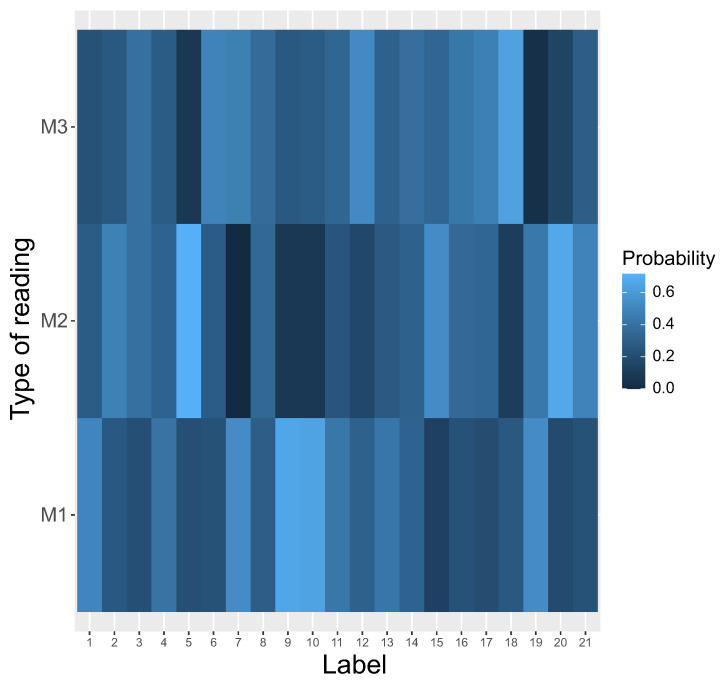
The heatmap of different types of reading systems (y-axis) at the beginning of simulation. This is a graphical representation of a matrix in which each genetic label ascribes a probability that is read by a given reading type.

**Figure 6 ijms-23-01690-f006:**
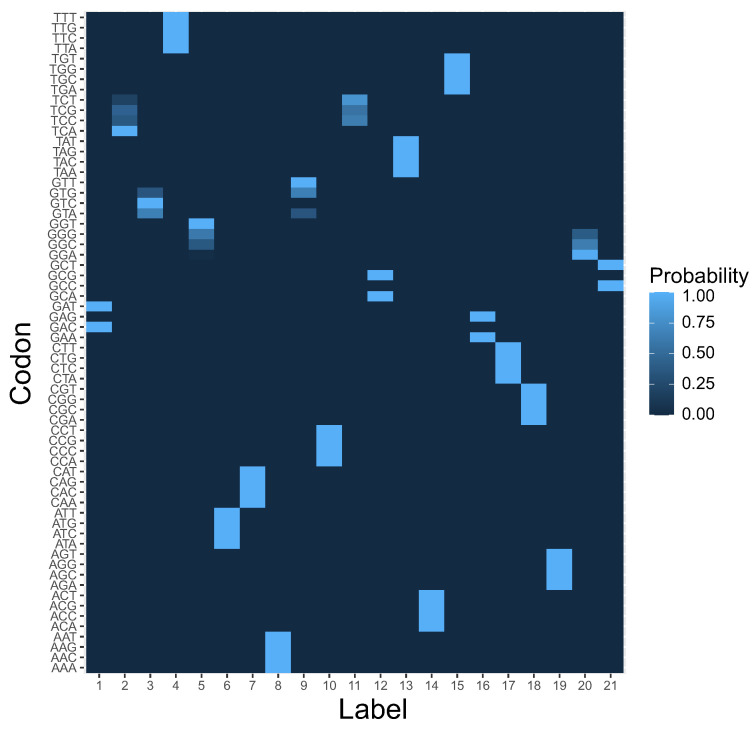
The heatmap of genetic code encoding 21 labels by 64 codons at the end of simulation. This is a graphical representation of the matrix $P=(p_{cl})$, in which each element $p_{cl}$ ascribes a probability that a codon *c* in a row encodes a label *l* in a column. Please compare with Figure 7 at the beginning of the simulations.

**Figure 7 ijms-23-01690-f007:**
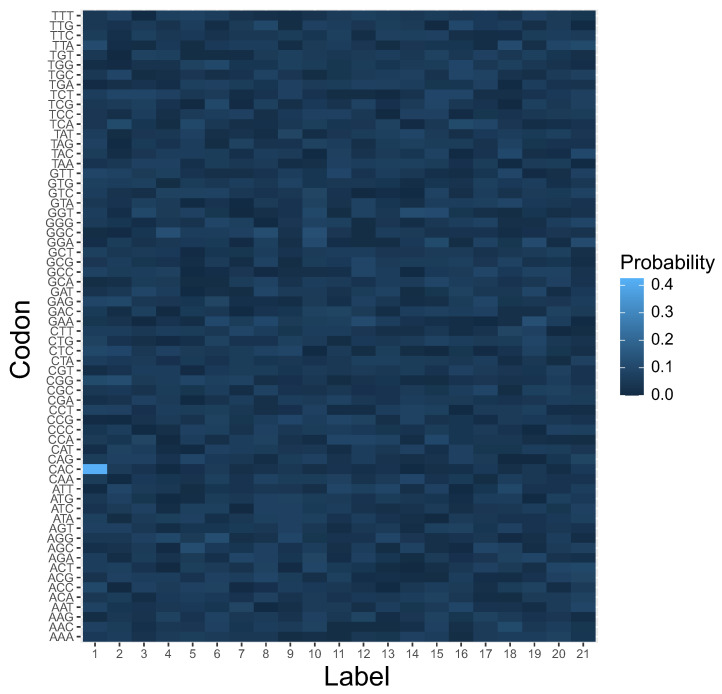
The heatmap of genetic code encoding 21 labels by 64 codons at the beginning of the simulation. This is a graphical representation of the matrix $P=(p_{cl})$, in which each element $p_{cl}$ ascribes a probability that a codon *c* in a row encodes a label *l* in a column.

**Figure 8 ijms-23-01690-f008:**
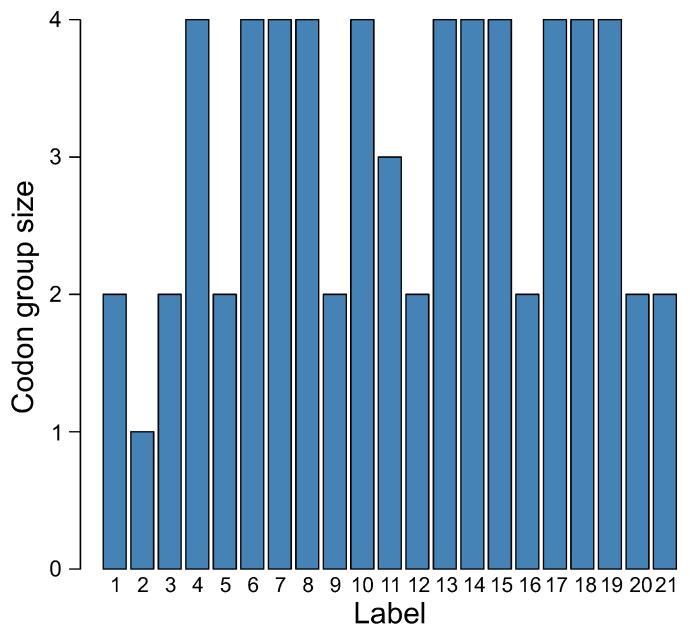
The distribution of the size of codon groups encoding a given label in the genetic code at the end of the simulation.

## Data Availability

The computations were conducted using C++ programming language. All source codes and raw data relevant to our investigations were included in Appendix A.

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
