# Peer review of "Model of Genetic Code Structure Evolution under Various Types of Codon Reading"

_ijms, 2022, doi:10.3390/ijms23031690_

Round 1

Reviewer 1 Report

Dear authors,

This article studies the potential emergence of genetic codes from the set of highly ambiguous codon assignments to the codes, carrying out a theoretical model of genetic code structure. The manuscript is well written and structured, is very novel, the introduction provides sufficient background and includes relevant references, the research design is appropriate, the methods are adequately described, the results are clearly presented, and the conclusions are supported by the results. I would like to congratulate the authors for the originality of the work presented, which, from my point of view, can be published in its current version.

Author Response

Thank you very much for the revision of our manuscript. We are really
glad that you found our manuscript interesting. We extended and improved
the discussion including additional interesting aspects about the
genetic code and translation mechanism.

Reviewer 2 Report

The manuscript submitted by BÅ‚azej and colleagues deals about the modelling of the genetic code evolution comparing three different scenarios. The topic of the submitted manuscript is highly interesting and the simulation reported is very intriguing and well planned/performed. In particular, Blazey et al studied the possible mechanisms of evolution of the standard genetic code by computer simulations showing that its evolution has been driven to evolve as a system resistant to errors/mutations and with low entropy.

The text is generally well written (I think that there isn’t any need of language revision) and in particular the introduction is really effective to introduce readers to the topic of the manuscript. Methods are adequately described and the same is true for the results that are clearly reported/described. The discussion is highly focused but very short and I think that it should be improved.

As a whole, I suggest that manuscript should be accepted after minor revision. In particular:

  1. I suggest the revision of the last part of the abstract in order to make more evident the Author’s results in respect to the state of the art;
  2. The discussion needs to be improved by explaining future research perspectives and I think that Authors should also take into account the possible inhibitory effect of changes in the code due to the advantages of horizontal gene transfer;
  3. Some published papers (e.g. Koonin and Novozhilov, doi: 1080/21541264.2021.1927652 cited in the submission) moving from the observation thattRNAomes (all the tRNAs of an organism) are simpler in archaeal systems relative to bacterial systems suggested that tRNA evolution occurred by duplication from a proto-tRNA molecules. Can Authors also include similar scenarios in their discussion? At the same time current tRNA could be the result of evolution after the establishment of the standard genetic code. Judging from the tRNA structure, cloverleaf tRNA appears to represent at least a second-generation scheme that replaced a robust 31-nt minihelix protein-coding system, evidence for which is preserved in the cloverleaf structure. Can Authors also include similar scenarios in their discussion?
  4. If Authors confirm their preference for a short discussion, I suggest to modify the section “Results” into a section “Results and discussion”, with the introduction of a final section related to Conclusions.

Author Response

Thank you very much for the revision of our manuscript. We are really
glad that you found our manuscript interesting. We corrected the
manuscript according to your remarks. We modified the second part of the
abstract in respect to the state of the art. Moreover, we extended and
improved the discussion including the advantages of horizontal gene
transfer, tRNA evolution by duplication of their genes, error
minimization by the genetic code and mistranslation errors.